# iADRGSE: A Graph-Embedding and Self-Attention Encoding for Identifying Adverse Drug Reaction in the Earlier Phase of Drug Development

**DOI:** 10.3390/ijms232416216

**Published:** 2022-12-19

**Authors:** Xiang Cheng, Meiling Cheng, Liyi Yu, Xuan Xiao

**Affiliations:** Department of Computer, Jingdezhen Ceramic University, Jingdezhen 333403, China

**Keywords:** adverse drug reactions, graph isomorphism network, self-attention, multi-label learning

## Abstract

Adverse drug reactions (ADRs) are a major issue to be addressed by the pharmaceutical industry. Early and accurate detection of potential ADRs contributes to enhancing drug safety and reducing financial expenses. The majority of the approaches that have been employed to identify ADRs are limited to determining whether a drug exhibits an ADR, rather than identifying the exact type of ADR. By introducing the “multi-level feature-fusion deep-learning model”, a new predictor, called iADRGSE, has been developed, which can be used to identify adverse drug reactions at the early stage of drug discovery. iADRGSE integrates a self-attentive module and a graph-network module that can extract one-dimensional sub-structure sequence information and two-dimensional chemical-structure graph information of drug molecules. As a demonstration, cross-validation and independent testing were performed with iADRGSE on a dataset of ADRs classified into 27 categories, based on SOC (system organ classification). In addition, experiments comparing iADRGSE with approaches such as NPF were conducted on the OMOP dataset, using the jackknife test method. Experiments show that iADRGSE was superior to existing state-of-the-art predictors.

## 1. Introduction

Adverse drug reactions (ADRs) or side effects are substantially harmful or distressing reactions, and are described as adverse responses to drugs beyond their anticipated therapeutic effects [1]. In the United States, it is estimated that ADRs result in over 100,000 patient deaths per year [2] and the cost of ADRs-related morbidity was USD 528.4 billion in 2016 [3]. The process of drug-development involves a lot of monetary resources because it involves a lot of clinical trials and tests [4]. Many ADRs are not detected in the early stages of drug development, owing to restricted trial samples and time [5]. Thus, ADRs not only jeopardize patient health but also result in wasted healthcare costs, and are considered as a major global public-health problem. Traditional laboratory experiments to identify potential ADRs are not merely cumbersome and low cost-effective, but also less effective in the earlier phase. In recent years, algorithms in silico have been employed to speed up the prediction process and reduce drug-development costs.

Among the existing studies, some utilize data mining to analyze potential ADRs from large amounts of data and various sources of information; others adopts machine learning methods to predict ADRs.

The available databases of ADRs have some limitations at present. The data collected by the spontaneous reporting systems (SRS) and FDA Adverse Event Reporting System (FAERS) are not comprehensive enough, and there are problems such as repeated declaration. Drugs in the Side Effect Resource (SIDER) are limited to FDA-approved drugs only. The content of the European Medicines Agency (EMA) and other large-scale databases is complicated, and has no special retrieval of ADRs, which cause a lot of inconvenience for the use of data. Considering the limitations of the existing database, some researchers have mined the relationship between drugs and ADRs from texts, including SRS (covering spontaneous reports of adverse drug-events by healthcare professionals or patients), clinical narratives written by healthcare professionals, and electronic health records where diagnostic records are stored [6]. Other valuable big-data sources include social media posts such as health-related tweets, blogs, and forums [7]. Jagannatha and Hong trained the recurrent-neural-network framework (RNN) at the sentence- and document-levels to extract medical events and their attributes from unstructured electronic health records, and revealed that all RNN models outperform the counterfactual regret minimization (CRF) models [8]. An RNN model based on bi-directional long short-term memory (BiLSTM) networks has been proposed to treat text in social media posts as a sequence of words, and two BiLSTM models (BiLSTM-M2 and BiLSTM-M3) initialized with pre-trained embeddings perform significantly better than the BiLSTM-M1 model using random initialized embedding, because the pre-trained word embeddings are more effective in capturing the semantic similarity of words [9]. Ding et al. adopted character embedding and word embedding and combined them via an embedding-level attention mechanism to permit the model to determine how much information was used from the character-level or word-level component [10]. Although the previous attention methods have obtained good results in predicting ADRs, they only extract the individual semantic information entailed in a single sentence representation. In order to capture the different semantic information represented by different parts of the sentence, Zhang et al. developed a multi-hop self-attention mechanism (MSAM) model, in which each attention step aims to obtain different attention weights for different segments, in an attempt to capture multifaceted semantic information for ADR detection [11]. A weighted online recurrent extreme-learning-machine (WOR-ELM) method has been exploited to discriminate the boundaries of adverse drug reactions mentioned in biomedical texts [12]. It can be concluded from the above studies that both LSTM and the gate recurrent unit (GRU) are valuable tools for extracting ADRs from textual data. However, the methods of mining the ADRs from the text can only be used after the drug has been introduced onto the market, and cannot be used for the drugs in the research process.

Machine learning methods used to identify ADRs can be divided into three categories: similarity-based, network-topology-based, and matrix-decomposition-based.

The similarity-based methods are based on the fact that similar drugs have similar properties. It has been recognized that drugs with similar chemical structures exhibit similar biological activities; similar drug targets induce similar signal-cascade reactions, so they have similar ADRs [13]. Zhang et al. proposed a new method of measuring drug–drug similarity named “linear neighborhood similarity” to predict potential ADRs of drugs [14]. The diversification of drug information can enhance the predictive capability of such methods. In addition to drug chemical structures, drug target proteins, and drug substituents, Zheng et al. also used drug treatment information to identify ADRs [15]. Seo et al. applied the similarity of single-nucleic-acid polymorphism, side-effect anatomical hierarchy, drug–drug interaction, and target, and finally achieved better results by integrating the four predictors, random forest, logic regression, XGBOOST, and naïve bayes, using neural networks [13]. Liang et al. used a novel computational framework based on a multi-view and multi-label learning method to construct important drug features to improve predictor accuracy [16]. These predictors are similar in the use of learning classification models, and the key difference lies in the vectorized representation of drugs and ADRs.

The associations between drugs and other entities in the above methods are not integrated into the vector, so useful information may be lost. For this reason, the network-based method is also used to predict ADRs, and the new ADRs are inferred from the constructed network. Emir established a structural similarity network of drug chemical formulas and an ADRs transmission network to predict the potential ADRs [17]. Knowledge graphs (KGs) and their embedding process have become a useful tool in recent years, as they can not only represent the rich relationships between entities, but also directly encode these complex relationships as vectors. Using KG embedding to vectorize drugs and other entities is expected to better characterize drugs and other nodes. Bean et al. constructed a KG with four nodes, and vectorized it using an adjacent matrix of drug nodes to predict ADRs [18]. Emir et al. used KG to unify heterogeneous data from multiple databases, and the prediction of ADRs was regarded as a multi-label classification [19]. Zhang et al. designed a novel knowledge-graph-embedding method based on the Word2Vec model, and constructed a logistic-regression classification model to detect potential ADRs [20].

The matrix-decomposition algorithm decomposes the adjacency matrix of drug-ADRs pairs into multiple matrixes, and reconstructs an adjacency matrix to identify new drug-ADRs pairs. Liu et al. proposed a method based on structural matrix-decomposition named as LP-SDA, which is a label communication framework that links the chemical structure of a drug with the FDA Adverse Event Reporting System to predict ADRs [21]. Timilsina et al. integrated the bipartite graph which expressed the drugs and ADRs’ interactive relationship and the drug–drug network, where the edges represent semantic similarity between drugs using a matrix factorization method and a diffusion-based model [22]. DivePred was developed by Xuan et al., based on non-negative matrix factorization using disease similarity, various drug features of drug chemical substructures, and drug target-protein domains [23].

In recent years, graph neural networks (GNN) have been widely applied in various fields, and focus on mining potential information from the network structure. GNN has demonstrated its outstanding capability in the representation of biomolecular structures and relationships between molecules, and has received wide attention in the life sciences [24]. Withnall et al. introduced attentional and limbic memory-schemes into an existing message-passing neural network framework, and the prediction performance of molecular properties has been improved [25]. Furthermore, self-attentive mechanisms are frequently utilized in the field of natural language processing and are capable of efficiently processing text sequence-data [26]. In training the DDI prediction model, Schwarz et al. found that the model with an attention mechanism performed better than deep-neural-network models without attention [27].

In the early stage of drug design, there is no other information except the chemical-structure information of the drug. If the above methods relied only on the molecular formula structure to predict ADRs, the performance was very poor. For example, Dey et al. achieved an AUC of only 0.72 when using only the chemical structure of the drug to predict side effects [28]. Inspired by the advantages of GNN and self-attentive mechanisms, we propose an ADR multi-label prediction model called iADRGSE, which includes a self-attentive module based on drug substructure sequences and a graph network module on drug chemical-structure maps. The structure of this dual-channel model can effectively adapt to the different structural information of drugs, and improve the ability to predict ADRs. To verify the performance of the model, we collected data from the adverse drug reaction classification system (ADRECS), and classified the types of ADRs into 27 categories, in accordance with system organ type (SOC). The iADRGSE demonstrated better performance than other state-of-the-art methods in a multi-label ADRs prediction task.

## 2. Results and Discussion

### 2.1. Evaluation Metrics

ADRs prediction is a multi-label classification problem. The quality of multi-label learning is evaluated as more complex than single-label classification, because each sample is a label set. The metrics such as accuracy, precision, recall, AUC, and AUPR are frequently used. The last four metrics set the parameter average = ‘macro’, which represents the average of the metrics independently calculated over the 27 labels. Their formulas are as follows:(1)$$Accuracy=\frac{1}{N}\overset{N}{\underset{j=1}{\sum }}\frac{\left(TP_{j}+TN_{j}\right)}{\left(TP_{j}+FN_{j}+FP_{j}+TN_{j}\right)}$$
(2)$$Precision\left(macro\right)=\frac{1}{L}\overset{L}{\underset{i=1}{\sum }}\left(\overset{N}{\underset{j=1}{\sum }}\frac{TP_{ij}}{TP_{ij}+FP_{ij}}\right)$$
(3)$$Recall\left(macro\right)=\frac{1}{L}\overset{L}{\underset{i=1}{\sum }}\left(\overset{N}{\underset{j=1}{\sum }}\frac{TP_{ij}}{TP_{ij}+FN_{ij}}\right)$$
where *TP*, *TN*, *FP*, and *NP* denote true positive, true negative, false positive, and false negative, respectively. *N* stands for the count of samples, and *L* represents the number of labels. Accuracy represents the proportion of drugs that are correctly predicted. Precision stands for the fraction of drugs that are predicted to be positive which is actually correct. Recall means the fraction of drugs that are truly labeled as positive which is correctly predicted; AUC is the area under the receiver operating characteristic curve; AUPR indicates the area under the precision recall curve.

### 2.2. Parameter Setting

We randomly selected 90% of the collated 2248 drugs as a training dataset for constructing and training the prediction model, and the remaining 10% as an independent-testing dataset, to test the constructed model. The selection of hyperparameter and feature-evaluation experiments were all optimized by a five-fold cross-validation test.

There are four parameters which have the greatest impact on the performance of the iADRGSE deep learning model: parameter h for the number of heads in attention, parameter ε for the dropout rate, parameter ξ for the learning rate in the model training, and parameter δ for the L2-regularization. It was observed from Figure 1 that when h = 2, ε = 0.5, ξ = 0.001, δ = 0.001, the performance reached its optimal value. Generally speaking, multiple heads are preferable to single heads, but more heads are not necessarily better. As shown in Figure 1a, the performance of the model is similar when the heads are set to 4 or 8, and the AUC is increased by 4.41% when the heads are 2. The effect of L2-regularization on the model is illustrated in Figure 1b, where the model works better with this hyperparameter of 0.001.

### 2.3. Feature Evaluation

We assessed the impact of various combinations of drug features on the forecasting of ADRs, and used the model’s metric scores as an indicator of the usability of the feature combinations. The results for different hierarchical feature combinations are displayed in Table 1.

In this study, the graph channel generated the mutual information, the edge information, and the context information of the drug molecule graph. We found that removal of one or two of the graph channel features had little effect on the performance of the model, but if the graph channel features were all removed, the performance of the model decreased significantly, and the AUC especially was reduced by 5%. Consequently, graph-embedding features are fairly significant in the model. In addition, in order to demonstrate that self-attentive coding of sequential channels can extract task-related features, we carried out experiments without encoders. The experiments demonstrated that the performance of the model without the encoders decreased in all metrics, and the AUC was reduced by close to 3%. We also compared feature-fusion methods by applying the mean value algorithm with our method, and the results revealed that our model feature- fusion method was more competitive.

### 2.4. Comparison of Feature-Extraction Methods of iADRGSE and Several Classic Feature-Extraction Methods

We compared the feature-extraction methods of iADRGSE with other feature-extraction methods, such as CNN_FP2, BERT_smiles [29], and Attentive_FP [30]; the hyperparameter settings for these three baseline methods are given in Appendix A. CNN is the most frequently used deep learning method in the field of vision. CNN_FP2 is the method of mining FP2 information through the convolution neural network. BERT_smiles use pre-trained bidirectional encoder representations from transformers (BERT) to extract SMILES sequence features. Attentive is the method of mining molecular fingerprint information though the graph attention-based approach. The results of the proposed iADRGSE predictor and the above three feature-extraction methods are also presented in Table 1. It is not difficult to find that the CNN_FP2 approach is better than the BERT_smiles and Attentive_FP. Moreover, the iADRGSE_no_Gin method, which also uses FP2 as input, remarkably outperforms CNN_FP2 by approximately 4% in AUC and AUPR. This demonstrates the superiority of sequence-based self-attentive encoding methods in feature extraction. The proposed iADRGSE predictor in this paper remarkably outperformed the above three feature-extraction methods in all metrics from Table 1, and outperformed CNN_FP2 by approximately 3.15% in accuracy, 6.86% in precision, 9.19% in AUC, and 7.23% in AUPR.

Independent tests can better verify the robustness of the prediction models. We also tested the performance of the iADRGSE and the above three feature-extraction methods using the independent test set, and the results are listed in Table 2. The predictive results of iADRGSE are very stable, at approximately 0.8196, 0.7632, 0.7461, 0.7735 and 0.7950 for accuracy, precision, recall, AUC and AUPR, respectively. It can be observed from the table that the accuracy score obtained by the current iADRGSE is significantly higher than that of the other three models, as are the other three indicators, except that recall is slightly lower than for BERT_smiles.

### 2.5. Comparison with Existing Predictor

Our model only used the chemical structure of the drug, which is helpful for the detection of ADRs in the preclinical stage of drug development. To further illustrate our approach, we compared the performance of iADRGSE with those of other models employing only chemical structures (NFP [28], circular fingerprinting [31]), and two drug-safety signal-detection algorithms (MGPS [32] and MCEM [33]), using the jackknife test method. For convenience of comparison, the scores of the five indexes obtained by these five predictors based on the OMOP dataset are listed in Figure 2. It can be observed from the figure that the AUC obtained by the iADRGSE model is significantly higher than that of the existing predictors, and remarkably outperforms the finest result of the comparison method by approximately 7%, in AUC.

In addition, using only the chemical-structure information of the drug, our model achieved good performance on the drug-safety signal-detection task (AUC = 0.7877), which provides a favorable complementary approach for toxicity detection in the early stages of drug design.

### 2.6. Case study

In this section, we undertake a case study to demonstrate the usability of iADRGSE in practice. In accordance with the loss value, the top 100 drugs were selected for case analysis to verify the ability of the model to predict potential ADRs. Next, comparing the predicted results of these 100 drugs with the true values, we found 21 drugs with potential adverse effects, whose predicted values are found in Appendix A. These 21 drugs had a total of 23 pairs of potential adverse reactions, as shown in Figure 3. Finally, we analyzed the predicted results in detail, and mainly focused on 23 pairs of potential adverse reactions, in Table 3.

For these 23 pairs of potential adverse reactions, we applied the search tools provided by medthority.com (accessed on 2 December 2022), reports from clinicaltrial.gov, and the related literature in PubMed and et al., to find the supporting evidence for them. From Table 3, we can observe that 17 of the 23 pairs of potential adverse reactions have evidence for them, indicating that the accuracy of the model iADRGSE has been further improved. For instance, the drug Pomalidomide carries a risk of social circumstances, which was reported in the literature [34].

## 3. Materials and Methods

### 3.1. Dataset

ADRECS [35] is an adverse-drug-reaction database that contains 2526 drugs and 9375 types of ADRs. To guarantee the quality, the drugs data were screened strictly according to the following criteria: (1) drugs without PubChem_ID were removed because PubChem_ID should be used to acquire the drug SMILES in the PubChem database; (2) drugs having no SMILES were removed. After following strictly the above two procedures, we finally obtained an ADRs dataset that contained 2248 drugs. We classified the 9375 adverse-drug-reaction types into 27 categories according to system organ classification (SOC). Finally, we obtained 2248 drugs, of which 27 belong to one ADR attribute, 32 to two different ADR attributes and so on. Detailed information is shown in Figure 4.

For other details of the dataset, please see Figure 5 and Figure 6. It is apparent that the data is unbalanced. The label base and density of the dataset are 16.5 and 0.6111, respectively. The base and density are relevant to the learning hardness of the multi-label classifier, i.e., the lower the density and the larger the base, the more difficult the multi-label learning process [36]. The dataset can be downloaded from the website https://github.com/cathrienli/iADRGSE (accessed on 10 December 2022).

In this study, the same dataset as that investigated in Harpaz et al. [37] was adopted for demonstration. The reason we chose it as a comparison dataset for the current study is that the OMOP dataset is derived from real-world data, such as the FDA Adverse Event Reporting System (FAERS) and data reported in recent papers. This dataset consisted of 171 drugs and four ADRs (acute kidney injury, acute liver injury, acute myocardial infarction, and gastrointestinal bleeding). Dataset statistics are provided in Table 4.

### 3.2. Problem Formulation

The core of our work is to construct a one-to-many mapping $F:d_{i}\to {\left\{l_{ij}\right\}}_{j=1}^{Nl}$ between a set of drugs $D=\left\{d_{i}|1\leq i\leq N_{d}\right\}$ and a set of ADR labels $L=\left\{l_{j}|l_{j}\in \left[0,1\right],\;1\leq j\leq N_{l}\right\}$, where $N_{d}$ is the number of drugs and $N_{l}$ represents the number of labels. For the multi-label ADR task, we define the label $l_{j}=1$ if the drug belongs to the j-th ADR class; otherwise, $l_{j}=0$. In this study, each drug is expressed by two parts: molecular structure maps and substructure sequences.

### 3.3. Overview of iADRGSE

The system architecture of iADRGSE is shown in Figure 7. The architecture can be divided into feature extraction and prediction modules. In the feature-extraction module, a dual-channel network with sequence channel and graph channel is constructed, to learn the various structural features of drugs. In the graph channel, drug molecules are represented as chemical structure graphs, and we use the pre-trained graph isomorphism network (GIN) [38] to obtain various physicochemical properties of drugs. The sequence channel is connected by three units of preprocessing, encoder and feedforward in tandem, which aims to extract the substructural features of drug molecules. In the preprocessing unit, word embedding is applied to generate dense vectors from drug substructure sequences, and these vectors are fed to a downstream module for feature mining. The encoder mainly utilizes the multi-head self-attention mechanism from the transformer [26] network to perform a weighted combination of substructure embeddings. The feedforward unit reduces the dimensionality of the encoded features to adapt to the subsequent prediction task. Finally, in the prediction module, we concatenate diverse structural features learned from the upstream phase and input them into the deep neural networks (DNN) to predict the ADR labels.

### 3.4. Drug Molecular Representation

The simplified molecular-input line entry system (SMILES) is a specification in the form of a line notation for describing the structure of chemical species, using short ASCII strings [35]. We invoked the RDKit [36] tool to convert the SMILES of the drug, $d_{i}$, into a molecular structure graph $g_{i}=\left(D,E\right)$, where node set D represents atoms and edge set E represents chemical bonds. Node information carries atomic attributes such as atom type, atomic number, atom degree, electrons, hybridization, aromatic, etc. Edge information involves bond type, conjugated, ring, etc. Inspired by MUFFIN [37], in this study, we adopted the information of the number and chirality of the atom and the type and direction of the bond.

The FP2 fingerprint format Is a path-based fingerprint, which can be generated by Open Babel [38]. FP2 can represent drugs as 1024-dimensional binary vectors according to chemical substructures, with each dimension indicating the presence or absence of the corresponding substructure. To avoid sparsity, the drug representation thus obtained is a 256-digit hexadecimal string.

### 3.5. Feature Learning

#### 3.5.1. Graph Channel

In order to parse graph-structured data, the GIN model is used because of its powerful function in the field of the graph neural network. The GIN model is pre-trained on individual nodes as well as the entire graph, to learn local and global representations. The iADRGSE model applies the pre-trained models of deep-graph information maximum [39] (Infomax), raw edge-prediction [40] (Edge), and context prediction [41] (Context), based on the GIN architecture, to generate the mutual information, $X^{m}$, the edge information, $X^{e}$, and the context information, $X^{c}$, of the drug molecule graph, respectively.

The information-extraction process includes a message-passing stage and a readout stage. The message passing is to conduct the aggregation function, $M_{t}$, to collect the information of neighboring nodes and edges, and to fuse the aggregation information to the current node through the update function, $U_{t}$. Therefore, message passing can be described as below:(4)$$m_{u}^{t+1}=\underset{v\epsilon N\left(u\right)}{\sum }M_{t}\left(h_{u}^{t},\;h_{v}^{t},\;e_{uv}\right)$$
(5)$$h_{u}^{t+1}=U_{t}\left(h_{u}^{t},m_{u}^{t+1}\right)\;$$
where t is the number of iterations, u denotes the *u*th node in the graph, $g_{i}$, $N\left(u\right)$ represents the adjacent nodes of node u, $h_{u}^{t}$ stands for the intermediate state of node u at time, t, $e_{uv}\;\epsilon \;E$ indicates the attributes of edge between u and v. In particular, both $M_{t}$ and $U_{t}$ are inherently linear layers with the same dimension weight matrix W, which have been used to transfer information between nodes. Finally, the eigenvector, $X_{i}$, of graph $g_{i}$ is calculated by MaxPooling on the node representation at the t step. The readout phrase can be formulated as:(6)$$X_{i}^{*}=MaxPooling(h_{u}^{t}\;|\;u\;\in \;g_{i}),*\in \left[m,e,c\right]$$

#### 3.5.2. Sequence Channel

Taking into account the sparsity of the substructure sequences, a word-embedding layer is used to preprocess the sequences. The substructure sequence is a hexadecimal string in which each of four substructures is represented by a hexadecimal digit. The word-embedding module assigns a dense learnable embedding-representation for each hexadecimal number, which are stored in a simple lookup table. The model retrieves the homologous word-embedding in accordance with the index (i.e., hexadecimal number) of the substructure. The layer-normalization [42] module re-standardizes the substructure-embedding vectors using the mean, μ, and variance, σ2, across the embedding dimension. Scale, γ, and bias, β, are learnable affine-transformation parameters and ϵ is a value added to the denominator for numerical stability. Therefore, the preprocessing feature, Ei, of the drug, di, is calculated as follows:(7)Ei=Embeddingqi−μσ2+ϵ*γ+β
where Ei ∈ ℝ256×dim′, dim′ is the word-embedding dimension.

To explore the different types of relevance that may exist between molecular substructures, we employ a multi-head self-attention mechanism, consisting of multiple parallel self-attention layers to encode substructure sequences. Each input vector, Ei,s, can be calculated out three new vectors, Qi,s, Ki,s, and  Vi,s based on three different linear transformation matrices, Wquery , Wkey,  and  Wvalue, respectively:(8)Qi,s|Ki,s|Vi,s=Ei,sWquery|Wkey|Wvalue
where s indexes the *s*th substructure embeddings in Ei, s ∈ 0,…,255, [Qi, s,Ki, s,Vi, s] ∈ ℝ1×dv, [Wquery, Wkey, Wvalue]∈ℝdim′×dv, dv is the vector dimension. Based on the aforementioned three intermediate vector matrices, we perform an attention-weighted sum over all substructures. The attention scores refer to the influence coefficients between the upstream and downstream substructures of the sequence, and are computed by the scaled dot-product of each substructure vector. For each substructure, s, the attention score, αi,s,j, of it and the substructure j∈0,…,255 can be calculated as follows:(9)αi,s,j=SoftmaxQi,sTKi,jdv

These attention scores, $\alpha_{i,\left(s,j\right)}$, and the vector, $V_{i,j}$, are weighted sum to generate a new vector, $O_{i,s}$, to represent the substructure, s. All the substructure vectors are simultaneously operated in parallel to obtain the new latent feature, $O_{i}$, of the drug, $d_{i}$.
(10)$$O_{i,s}=\overset{255}{\underset{j=0}{\sum }}\alpha_{i,\left(s,j\right)}V_{i,s}$$
(11)$$O_{i}=\left\{O_{i,0},..,O_{i,s},..,O_{i,255}\right\}$$

For the head number of self-attention H>1, the model actually runs the single-head self-attention function with H times based on the different parameter matrices Wqueryh,Wkeyh,Wvalueh in parallel, and a new feature representation, Oih, of the drug can be acquired, based on the *h*th self-attention head. These output values are concatenated and once again linearly transformed by the parameter matric Wo ∈ ℝhdv×dim′ to obtain output Oi ∈ ℝ256×dim′. The multi-head process is depicted as:(12)Oi=concatOi1,..,Oih,..,OiHWo

Note that we set $d_{v}=dim^{\prime }∕H$. To avoid the gradient problem, we add residual connection [43] to the input and output of the multi-head self-attention layer. The connection trick is to element-wise sum the output, $E_{i}$, of the previous preprocessing unit and the output, $O_{i}$, of the current multi-head self-attention layer. Finally, the residual features are transmitted through a layer-normalization module.

The fully connected feedforward-network consists of two linear layers, a batch-normalization [44] layer and a non-linear activation function, in order to further abstract and compress the latent encoded representation from the previous unit. Note that the output, $O_{i},\;$of the encoder unit is flattened before the linear transformation. The algorithm of batch normalization is the same as layer normalization, and the difference lies in which dimension is biased. The mean and standard-deviation are calculated per dimension, over the mini-batches. Ultimately, the substructure feature representation of the output of the sequence channel can be formulated as follows:(13)$$X_{i}^{f}=LeakyReLU\left(BN\left(Flatten\left(O_{i}\right)W_{layer1}\right)\right)W_{layer2}$$

#### 3.5.3. Multi-Label Classification

We spliced four structural features as drug representation, including mutual information Xim, edge information Xie, context information Xic and substructure information Xif. That is, drug di can be marked as:(14)Xi=concatXim,Xie,Xic,Xif
where Xi∈ℝ4dim, dim denotes the dimension of each structural feature.

Subsequently, Xi is fed into a single-layer linear network parameterized by Wpred, and an activation function is employed to output a predicted probability vector, Pi, where each component is deemed as the likelihood of a label. The process can be defined as follows:(15)Pi= σXiWpred
where Wpred ϵ ℝ4dim×Nl, σ refers to the sigmoid function for each Pi component.

## 4. Conclusions

In this study, we design a fast and effective prediction framework based on the fusion of graph embedding and self-attentive encoder features, named iADRGSE, to predict ADRs. Based on feature analysis and various kinds of experiments, the robustness and performance of iADRGSE is testified. The case study is conducted, in which the top 100 drugs are selected for analysis, and the study demonstrates that the model is competent in predicting the potential ADRs. For practical applications, a user-friendly online web server for iADRGSE is built at http://121.36.221.79/iADRGSE (accessed on 10 December 2022), which allows users to easily obtain results and brings great convenience to researchers.

iADRGSE obtains a better prediction performance than that of pervious methods. The primary reason is that iADRGSE fuses the graph-embedding and self-attentive encoder features of the drug, and these features are closely related to the prediction of ADRs.

It is anticipated that predictor iADRGSE will become a very useful tool for predicting ADRs at the early stage of drug discovery.

## Figures and Tables

**Figure 1 ijms-23-16216-f001:**
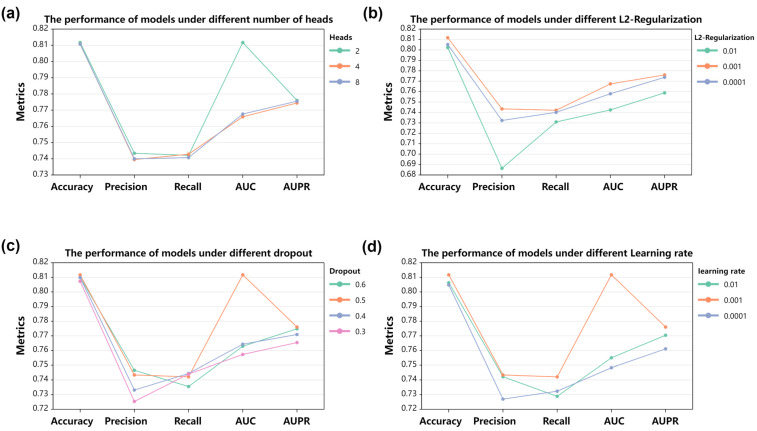
Model performance with different hyperparameter settings. (**a**)The performance of models under different number of heads. (**b**) The performance of models under different L2-Regularization. (**c**) The performance of models under different dropout. (**d**) The performance of models under different Learning rate.

**Figure 2 ijms-23-16216-f002:**
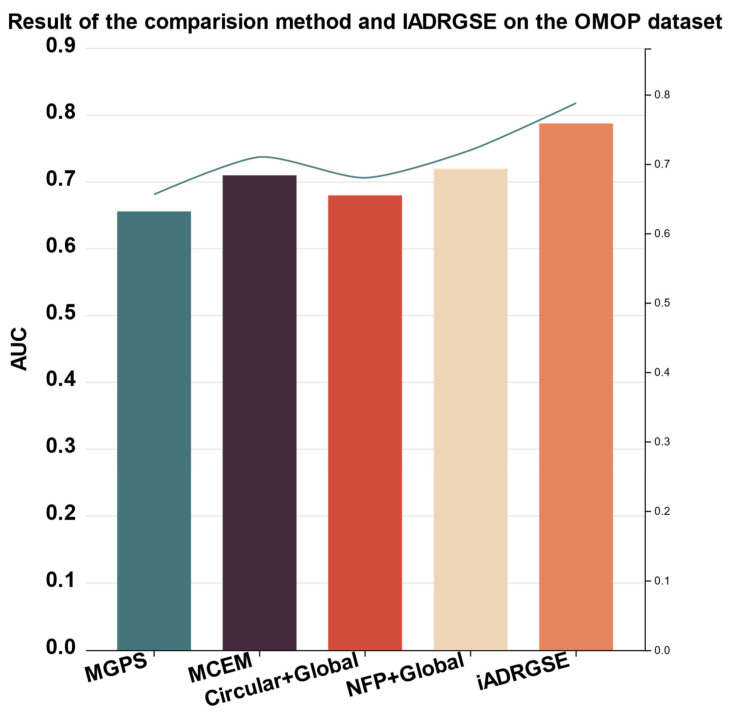
Results of the comparison method and iADRGSE on the OMOP dataset.

**Figure 3 ijms-23-16216-f003:**
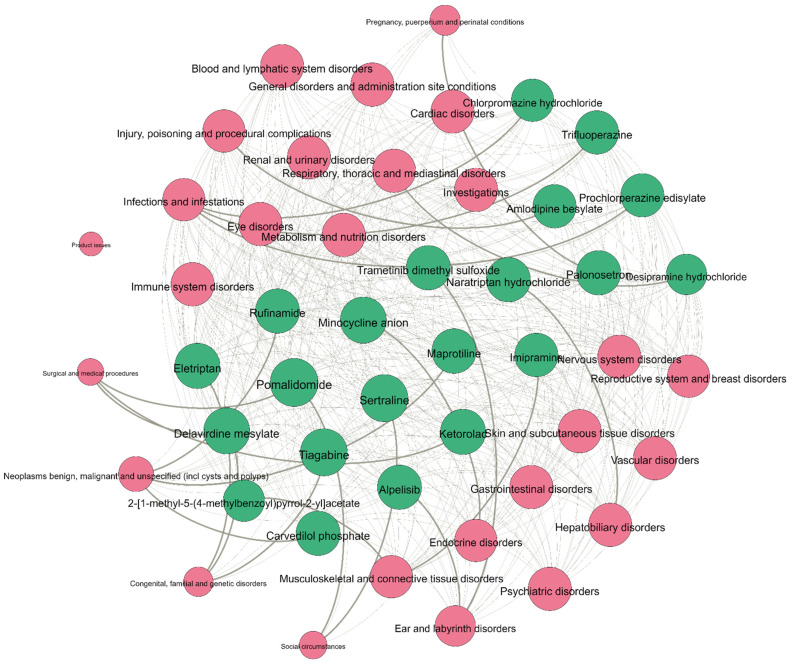
Drug and ADR association graph, 
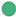
 represents the drug node, while 
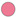
 represents the adverse reaction node; the more drugs connected to the ADR, the larger the node of the ADR; the line connecting the drug node and the ADR node indicates that the drug has these adverse reactions, and the thickened line indicates the potential adverse reactions of the drug.

**Figure 4 ijms-23-16216-f004:**
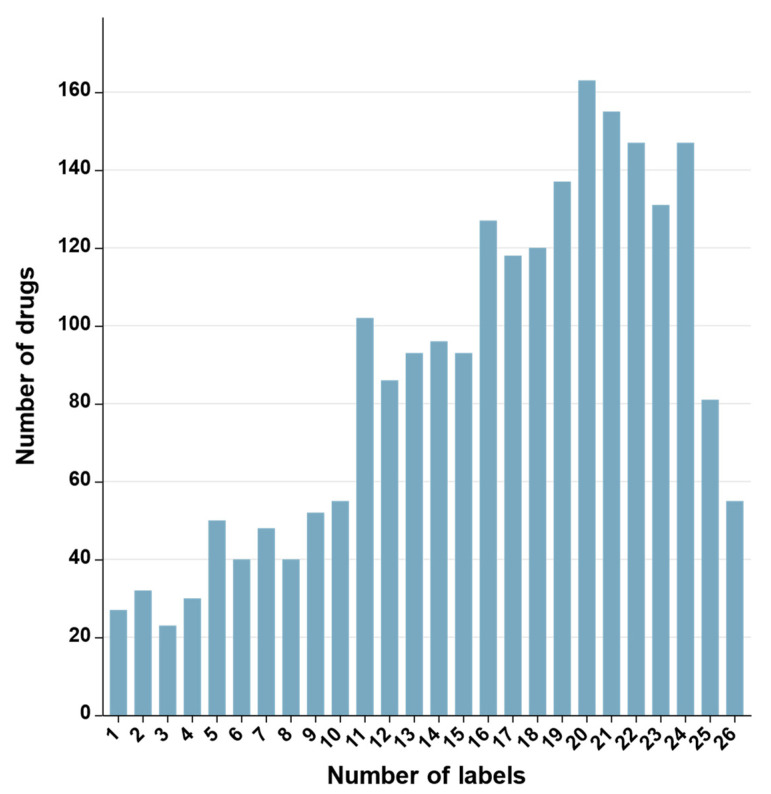
Number of drugs with one or more ADR types in the ADRs benchmark dataset.

**Figure 5 ijms-23-16216-f005:**
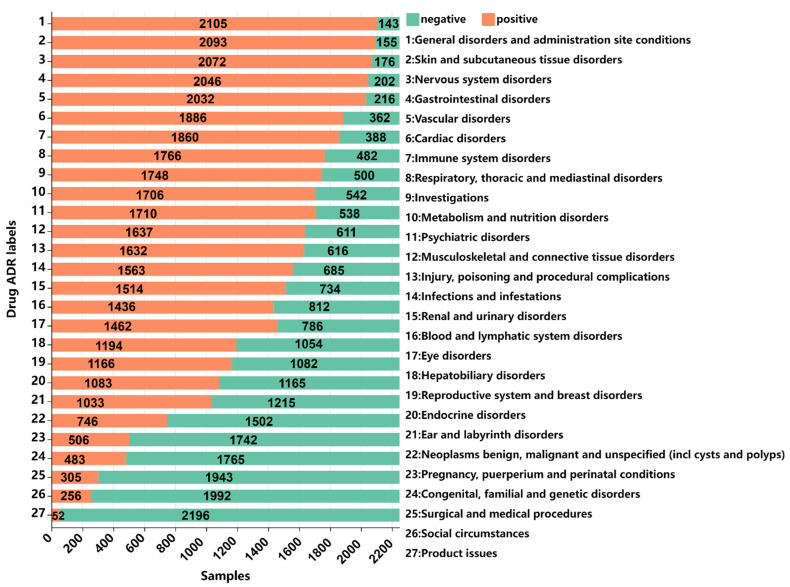
Sample distribution plot: horizontal axis represents the sample size, vertical axis represents the 27 ADR labels; 
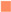
 represents positive samples, while 
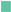
 is negative samples.

**Figure 6 ijms-23-16216-f006:**
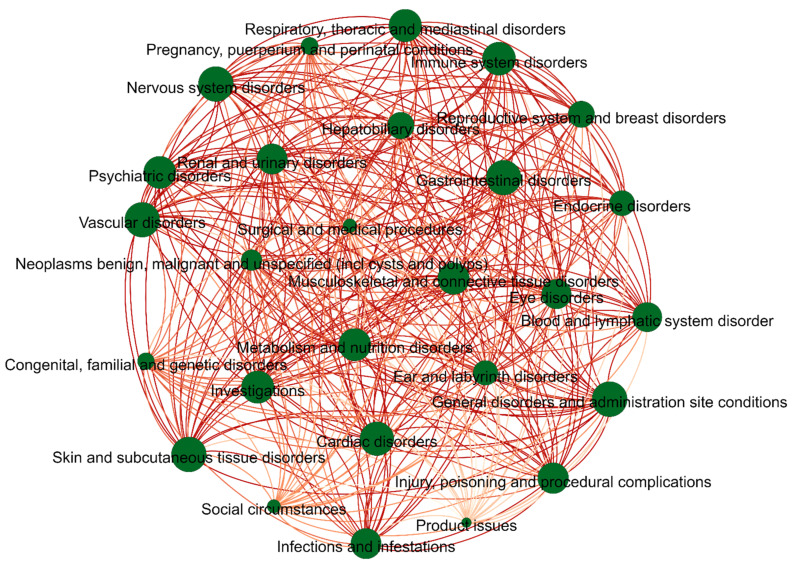
Label co-occurrence diagram, referring to the simultaneous occurrence of two labels; the green circle represents the label, and the size of the circle is the frequency of that ADR label; the red line connecting the two circles represents the simultaneous occurrence of these two ADR labels; the color shade of the edges indicates the number of times this group of labels appears; the darker the color, the more often this group of labels appears.

**Figure 7 ijms-23-16216-f007:**
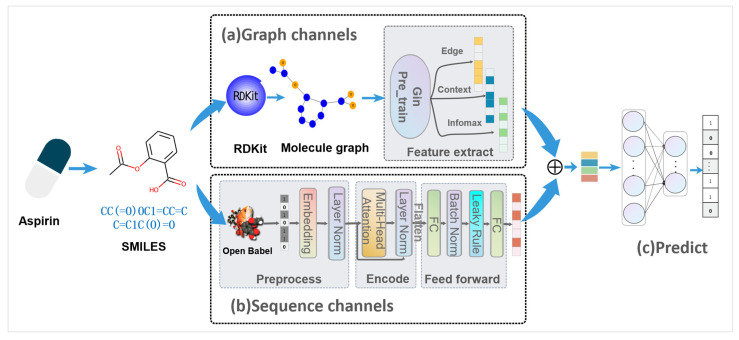
iADRGSE framework. (**a**) Graph channel. We perform the RDKit tool to convert the drug SMILES into chemical structure graphs and feed them into a pretrained GIN network, to learn graph-based structural information. (**b**) Sequence channel. The preprocessing unit utilizes Open Babel software to generate molecular substructure sequences from the SMILES of drugs. Then, the substructure sequences are represented as one-dimensional sequence vectors through the embedding layer. Next, the correlation information of each substructure is extracted further, using the encoder unit with a multi-head self-attention mechanism. Finally, the feed-forward unit (a multi-fully connected layer) receives encoded data from the upper layer to obtain the final sequence-based structural information of drugs. (**c**) Prediction module. These two types of structural information are concatenated and then mapped to the size of the labels, through an affine transformation for multi-label prediction.

**Table 1 ijms-23-16216-t001:** Results for different combinations of baseline, iADRGSE, and iADRGSE features.

Features Set	Accuracy	Precision (Macro)	Recall (Macro)	AUC (Macro)	AUPR (Macro)
CNN_FP2	0.7802 ± 0.0089	0.6474 ± 0.0213	0.7255 ± 0.0125	0.6726 ± 0.0145	0.7037 ± 0.0156
BERT_smiles	0.7754 ± 0.0084	0.6266 ± 0.0251	0.7246 ± 0.0091	0.6587 ± 0.0202	0.6987 ± 0.0150
Attentive_FP	0.7638 ± 0.0099	0.6748 ± 0.0130	0.7431 ± 0.0153	0.5669 ± 0.0234	0.6362 ± 0.0137
E + S	0.8074 ± 0.0083	0.7241 ± 0.0280	0.7350 ± 0.0145	0.7519 ± 0.0210	0.7590 ± 0.0156
C + S	0.8008 ± 0.0071	0.7251 ± 0.0120	0.7342 ± 0.0260	0.7545 ± 0.0163	0.7605 ± 0.0810
I + S	0.8044 ± 0.0081	0.7136 ± 0.0286	0.7282 ± 0.0172	0.7479 ± 0.0172	0.7533 ± 0.0172
E + C + S	0.8100 ± 0.0081	0.7369 ± 0.0256	0.7384 ± 0.0136	0.7628 ± 0.0148	0.7709 ± 0.0135
E + I + S	0.8078 ± 0.0081	0.7251 ± 0.0234	0.7342 ± 0.0138	0.7545 ± 0.0181	0.7605 ± 0.0138
C + I + S	0.8065 ± 0.0083	0.7245 ± 0.0305	0.7393 ± 0.0123	0.7580 ± 0.0125	0.7682 ± 0.0145
iADRGSE_Gin	0.7992 ± 0.0022	0.7450 ± 0.0103	0.7235 ± 0.0063	0.7358 ± 0.0113	0.7526 ± 0.0088
iADRGSE_no_Gin	0.7900 ± 0.0057	0.6888 ± 0.0323	**0.7506 ± 0.0176**	0.7098 ± 0.0179	0.7428 ± 0.0120
iADRGSE_ no_attention	0.8028 ± 0.011	**0.7451 ± 0.025**7	0.7302 ± 0.0117	0.7410 ± 0.0192	0.7619 ± 0.0139
iADRGSE_mean	0.7938 ± 0.0062	0.6793 ± 0.0352	0.7441 ± 0.0132	0.7206 ± 0.0161	0.7426 ± 0.0156
iADRGSE (ours)	**0.8117 ± 0.0089**	0.7434 ± 0.0266	0.7421 ± 0.0105	**0.7674 ± 0.0147**	**0.7760 ± 0.0130**

Note: E: Gin_Edge; C: Gin_Context; I: Gin_Infomax; S: based on sequence channel. iADRGSE_Gin: no sequence channel; iADRGSE_no_Gin: no graph channels; iADRGSE_no_attention: sequence channel has no self-attention; iADRGSE_mean: use the mean operation to fuse features. The best performance for each metric is shown in bold.

**Table 2 ijms-23-16216-t002:** Results of the baseline and iADRGSE on independent test sets.

Features Set	Accuracy	Precision (Macro)	Recall (Macro)	AUC (Macro)	AUPR (Macro)
CNN_FP2	0.8021	0.6960	0.7391	0.6990	0.7566
BERT_smiles	0.7949	0.6436	0.7523	0.6547	0.7196
Attentive_FP	0.7794	0.5791	0.7314	0.5398	0.6507
iADRGSE	**0.8196**	**0.7632**	**0.7461**	**0.7735**	**0.7950**

**Table 3 ijms-23-16216-t003:** Potential adverse-drug-reactions.

Drug Name	ADR	Evidence
Pomalidomide	Surgical and medical procedures	clinicaltrials.gov (all accessed on 2 December 2022)
Pomalidomide	Social circumstances	PMID: 35085238
Ketorolac	Surgical and medical procedures	cdek.liu.edu
Prochlorperazine edisylate	Infections and infestations	baxter.ca
Trametinib dimethyl sulfoxide	Ear and labyrinth disorders	clinicaltrials.gov
Trifluoperazine	Infections and infestations	healthline.com
Desipramine hydrochloride	Respiratory, thoracic and mediastinal disorders	rxlist.com
Chlorpromazine hydrochloride	Infections and infestations	Unconfirmed
Eletriptan	Congenital, familial and genetic disorders	Unconfirmed
2-[1-methyl-5-(4-methylbenzoyl) 3-pyrrol-2-yl]acetate	Musculoskeletal and connective-tissue disorders	medthority.com
Alpelisib	Ear and labyrinth disorders	clinicaltrials.gov
Imipramine	Musculoskeletal and connective-tissue disorders	cchr.org.au
Delavirdine mesylate	Congenital, familial and genetic disorders	rochecanada.com
Delavirdine mesylate	Delavirdine mesylate	Unconfirmed
Tiagabine	Congenital, familial and genetic disorders	Unconfirmed
Minocycline anion	Endocrine disorders	medthority.com
Naratriptan hydrochloride	Hepatobiliary disorders	medthority.com
Sertraline	Social circumstances	medthority.com
Amlodipine besylate	Injury, poisoning and procedural complications	PMID: 25097362
Palonosetron	Pregnancy, puerperium and perinatal conditions	Unconfirmed
Rufinamide	Neoplasms: benign, malignant and unspecified (incl cysts and polyps)	clinicaltrials.gov
Carvedilol phosphate	Neoplasms: benign, malignant and unspecified (incl cysts and polyps)	clinicaltrials.gov
Maprotiline	Neoplasms: benign, malignant and unspecified (incl cysts and polyps)	Unconfirmed

**Table 4 ijms-23-16216-t004:** Dataset statistics.

Datasets	Drug	ADRS Labels
ADRECS	2248	27
OMOP	171	4

## Data Availability

Not applicable.

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
