# Peer review of "iADRGSE: A Graph-Embedding and Self-Attention Encoding for Identifying Adverse Drug Reaction in the Earlier Phase of Drug Development"

_ijms, 2022, doi:10.3390/ijms232416216_

Round 1

Reviewer 1 Report

In this manuscript, the authors present a tool for predicting ADRs – iADRGSE. Tests show that the results obtained with this method are better than those previously used. I have no significant comments about the study. The description is clear, the methods are well-chosen. Looks good. I am not an expert in every aspect of this research, but based on my knowledge, I think the work deserves attention.

However, I have a little problem with the final decision. This manuscript, online server and application are a whole for me.  The authors claim that the web server is user-friendly and that the results are easy to obtain. Unfortunately, it seems to not be true. I tested the web server. I've done a few trials using the same drug. I only got the result once, almost instantly. Unfortunately, I have not received results from the other attempts, although it has been several days. Secondly, the result I got is completely incomprehensible. Simply table, in each column the same value 1.0. What does it mean? I do not know.

In my opinion, if the authors want to publish a work in which they promote iADRGSE, they should first prepare a good web server, i.e. improve the existing one. And not in reverse order. So, my decision is “minor”, because I hope that in a short time the authors can improve not so much the manuscript as the web server.

Author Response

To Reviewer #1

In this manuscript, the authors present a tool for predicting ADRs – iADRGSE. Tests show that the results obtained with this method are better than those previously used. I have no significant comments about the study. The description is clear, the methods are well-chosen. Looks good. I am not an expert in every aspect of this research, but based on my knowledge, I think the work deserves attention.

Response: We very much appreciate the overall positive comment of Reviewer #1.

However, I have a little problem with the final decision. This manuscript, online server and application are a whole for me.  The authors claim that the web server is user-friendly and that the results are easy to obtain. Unfortunately, it seems to not be true. I tested the web server. I've done a few trials using the same drug. I only got the result once, almost instantly. Unfortunately, I have not received results from the other attempts, although it has been several days. Secondly, the result I got is completely incomprehensible. Simply table, in each column the same value 1.0. What does it mean? I do not know.

In my opinion, if the authors want to publish a work in which they promote iADRGSE, they should first prepare a good web server, i.e. improve the existing one. And not in reverse order. So, my decision is “minor”, because I hope that in a short time the authors can improve not so much the manuscript as the web server.

Response: Thank you for your carefulness. We have carefully examined the article and corrected the errors. Due to the school power failure, the server stopped working, we forgot to restart the server after power on, this is our negligence, sorry. we have arranged personnel to check the server every day to make the server work normally. We followed your suggestion and have improved the output of the predicted results. If the drug has some side effect, it will show Yes, and if there is no such side effect, it will show No.

Reviewer 2 Report

The present manuscript shows scientific relevant results using a new data analysis tool, iADRGSE, to get information about adverse drug reaction in the early stages of drug design. It shows relevant improvements compared with currently used software. The manuscript is suitable to be accepted for publication, only minor points must be addressed before publication.

Minor points:

1.     Please provide all manuscript figures with better quality. For instance, in Figure 3 is hard to catch the differences in darkness of the color shade of the edges.

2.     Please check if the equation in line 180 is correctly displayed.

3.     Please check and correct typographic mistakes, there are several punctuation marks missed along the manuscript, for instance, in line 187 a period is needed after “Figure 4”.

Author Response

To Reviewer #2

The present manuscript shows scientific relevant results using a new data analysis tool, iADRGSE, to get information about adverse drug reaction in the early stages of drug design. It shows relevant improvements compared with currently used software. The manuscript is suitable to be accepted for publication, only minor points must be addressed before publication.

Response: We are very grateful for the reviewer to acknowledge the novelty, quality and significance of our work. We are deeply encouraged by reviewer’s positive comments and have made extensive efforts to address all the concerns.

Minor points:

  1. Please provide all manuscript figures with better quality. For instance, in Figure 3 is hard to catch the differences in darkness of the color shade of the edges.

Response: Thank you for your valuable advice! We followed the reviewer’s suggestion and provided all manuscript figures with better quality. Please see revised file.

  1. Please check if the equation in line 180 is correctly displayed.

Response: We very honor the circumspection and professional spirit of Reviewer #2. Corrections have been made. Please see the equation in line 183.

  1. Please check and correct typographic mistakes, there are several punctuation marks missed along the manuscript, for instance, in line 187 a period is needed after “Figure 4”.

Response: Thank you for your carefulness! We have carefully examined the article and corrected typographic mistakes.
